# The multi-faceted effects of technology-driven productivity surge in the crop & livestock sector in Greece: Evidence from the FABLE Calculator

Phoebe Koundouri[1,2,3,4], Konstantinos Dellis[1,4]*, Olympia Miziaki[5]

1 ReSEES Research Laboratory, Athens University of Economics and Business, Athens, Greece,
2 ATHENA RC Sustainable Development Unit, Athens, Greece, 3 Department of Earth Sciences and Peterhouse, University of Cambridge, Cambridge, United Kingdom, 4 UN Sustainable Development Solutions Network (SDSN) Global Climate Hub, Athens, Greece, 5 National Technical University of Athens, Athens, Greece

* kdellis@aueb.gr

## Abstract

This paper the effects of a technology-driven increase in crop and livestock productivity on key agricultural, land-use, and environmental indicators in Greece, using the FABLE (Food, Agriculture, Biodiversity, Land Use, and Energy) Calculator. Through empirical evidence and sophisticated modelling techniques, we analyze the intricate interplay between agricultural productivity and environmental sustainability. Our scenario-based projections show that higher agricultural productivity substantially reduces greenhouse gas emissions, primarily through lower livestock emissions, diminished pressure on pastureland, and increased emission withdrawals from land-use changes. Enhancing productivity in the livestock and crop sector reduces GHG emissions from agriculture by 29% until 2030 and 62% until 2050, compared to a business-as-usual scenario. The result is amplified when we embed the productivity surge in a holistic transformational strategy following Greece's national commitment including a shift to healthy dietary consumption. Moreover, costs decline markedly, by almost 50% in the long run, driven mainly by the reduction in pesticide use. In addition to its empirical findings, this paper delineates policy recommendations to support cutting-edge technologies within the Greek agricultural sector, focusing on horizontal and vertical measures. We highlight key precision agriculture technologies that align with current trends in Greece, particularly in the areas of drone applications, advanced sensors, and variable rate technology, alongside innovations in precision livestock management. Overall, our findings demonstrate that boosting agricultural productivity can generate a double dividend—lower emissions and enhanced competitiveness—particularly when supported by holistic policy measures.

**Data availability statement:** Most relevant data has been provided in this manuscript's Supporting information files. The FABLE calculator can be found via Zenodo using the following URL (Zenodo): https://zenodo.org/records/14638582.

**Funding:** The author(s) received no specific funding for this work.

**Competing interests:** No Authors have competing interests.

## 1. Introduction

The agricultural sector globally faces the pressing challenge of reconciling productivity with environmental sustainability, particularly in the context of escalating climate change and biodiversity loss. This issue is especially acute in Greece, a country with a rich agricultural heritage and diverse ecosystems [1]. Agriculture is a pivotal component of Greece's economy, contributing approximately 4% to the national GDP and employing around 11% of the workforce [2]. Greece is recognized for its high-quality agricultural products, which hold substantial market shares both domestically and internationally. These exports are crucial for the country's trade balance, generating significant income and supporting economic stability [3]. In addition to its economic contributions, the agricultural sector plays a critical role in rural areas, offering employment opportunities and helping to curb urban migration.

However, the sector faces significant challenges, including stagnant productivity, low technology adoption, an aging workforce, and small average farm sizes [4,5]. Balancing traditional farming practices with modern sustainability goals further complicates this landscape, requiring innovative solutions to navigate these often-competing demands [6]. Beyond structural constraints, information frictions and learning dynamics play a central role in shaping technology diffusion. Evidence shows that irrigation technology adoption is strongly influenced by social learning, spatial spillovers, and the presence of extension services, highlighting that productivity-enhancing technologies do not diffuse automatically but require targeted information and knowledge networks [7].

A key solution to these challenges lies in the adoption of advanced technologies, particularly precision agriculture (henceforth PA). As defined by ISPA "Precision Agriculture is a management strategy that gathers, processes, and analyzes temporal, spatial, and individual plant and animal data and combines it with other information to support management decisions according to estimated variability for improved resource use efficiency, productivity, quality, profitability, and sustainability of agricultural production" [8]. PA utilizes a range of cutting-edge technologies—such as drones, remote sensing, soil and plant sensors, and advanced analytics—to optimize resource use, enhance productivity, and minimize environmental impact. For example, drones equipped with multispectral cameras offer real-time monitoring of crop health, enabling targeted interventions that reduce resource waste and boost yields [9]. Soil and plant sensors provide detailed insights into soil conditions and plant health, allowing for precise irrigation and fertilization that improves crop yields while conserving water [10,11]. These technologies are further enhanced when combined with indirect technologies such as Variable Rate Technology (VRT) and Automated Steering Systems (ASS), which refine the application of inputs and machinery operation, leading to increased efficiency and productivity [12,13].

In the livestock sector, technologies such as automated feeding and milking systems, along with wearable sensors, play a significant role in improving animal welfare and productivity. These systems adjust feed rations and monitor animal health in real-time, leading to better growth rates and increased efficiency [14–16]. Adoption

of productivity-enhancing technologies in agriculture is shaped by risk, uncertainty, and farmers' expectations. Despite the potential of these technologies, there remains a noticeable gap in the literature concerning comprehensive analyses that integrate technological innovations with sustainability in Greece's agricultural sector [17,18]. This study aims to bridge this gap by critically assessing the potential of key technologies to drive productivity gains in the Greek agricultural landscape. Few studies have explored the relationship between subpar technological development and the lagging performance of the Greek agricultural sector in the field of sustainable transformation. Land fragmentation and small holding size limit the economic viability of investing in mechanization and advanced technologies, resulting in low productivity and income for small farmers [19]. Kalfas et al use a stratified sampling technique to survey 240 Greek farmers to find that the adoption of agricultural technologies enhances sustainability [20]. Another study casts focus on a failed project trying to introduce Integrated Crop Management (ICM) in a Greek village [21]. The results underscores that the socioeconomic factors, including the lack of technological skills and innovation networks bridging scientific knowledge and application, rendered impossible the diffusion of technological knowledge to sustainable agricultural outcomes. In this study we tangibly assess the sustainability potential of enhanced productivity, especially for climate change mitigation, and offer insights into achieving this productivity surge.

To address this, the study focuses on three fundamental questions:

(i) How does the enhancement of crop and livestock productivity influence key Agricultural, Forestry, and Other Land Use (AFOLU) indicators in Greece, as quantified through the FABLE Calculator?

(ii) Which technological advancements are most suited to achieving increased agricultural productivity considering Greece's unique agricultural characteristics?

(iii) Which policy mechanisms are most effective in promoting the adoption and dissemination of cutting-edge agricultural technologies that are pivotal for enhancing productivity within the Greek agricultural sector?

Utilizing the FABLE (Food, Agriculture, Biodiversity, Land Use, and Energy) Calculator—a sophisticated tool for modeling and scenario analysis—this study provides empirical evidence on the outcomes of increased agricultural productivity under various scenarios. The FABLE Calculator offers a robust framework for modeling complex interactions, allowing for the development of mid-century projections for the Greek agri-food sector under different conditions, including a Business-as-Usual Scenario and a scenario integrated with a series of reforms aligned with Greece's national commitments.

Furthermore, the study explores the policies necessary to support the widespread adoption of these technologies in a manner that is efficient and equitable. The study advocates for policies that prioritize environmental sustainability and social equity, ensuring that the benefits of technological innovations are shared inclusively across all segments of society. This balanced approach is essential for achieving sustainable economic growth in Greece's agricultural sector, aligning technological advancements with broader sustainability objectives.

In summary, this research contributes to the scientific literature by offering a comprehensive analysis of the potential benefits of increasing agricultural productivity in Greece. In addition, we use data-driven scenario-based projection from the FABLE Calculator, a modelling tool developed specifically for the case of the Greek agri-food sector. Through the FABLE projections, we are able to evaluate the projected benefits of productivity-enhancing reforms on key environmental and socio-economic variables up to 2050 and contribute tangibly to the justification of these policy measures. While previous studies have examined technology adoption, productivity constraints, or sustainability outcomes in isolation, our analysis integrates empirically grounded productivity shocks for both crop and livestock systems within a coherent national modeling framework. This enables a simultaneous and internally consistent evaluation of greenhouse gas emissions, land-use dynamics, and production costs up to 2050 across alternative FABLE pathways. Crucially, the paper links these quantified outcomes directly to policy design: horizontal and vertical policy recommendations are not presented abstractly but are explicitly motivated and justified by pathway-specific results on emissions abatement, land sparing, and cost reductions. Through a comprehensive empirical and theoretical framework, it provides practical insights for policymakers

and stakeholders, emphasizing the importance of integrating technological innovations with socio-economic and environmental considerations. This integrated evidence-based approach provides a novel decision-support contribution for Greek agricultural and climate policy, distinguishing the present study from prior work that lacks comparable modeling depth, national specificity, or explicit policy–outcome linkages.

The structure of this paper is as follows: Section 2 discusses the FABLE approach to evaluating sustainability in agri-food systems and the methodological aspects of the FABLE Calculator; Section 3 outlines the basic projections using the FABLE Calculator; Section 4 provides horizontal and vertical policy recommendations; Section 5 discusses the results and their implications; and the final Section concludes.

## 2. Material and methods: The FABLE approach for developing land-use and climate scenarios

The FABLE Consortium is a global collaboration of researchers focused on developing national pathways aligned with global sustainability objectives, such as the Sustainable Development Goals (SDGs) and the Paris Climate Agreement. Researchers and stakeholders from various countries work together to develop strategies that are scientifically robust and politically feasible. These pathways address key factors such as agricultural productivity, biodiversity conservation, greenhouse gas emissions, and socio-economic development, emphasizing data-driven decision-making and the sharing of knowledge and best practices across countries. Through international collaboration and integrated modeling, FABLE aims to guide policymakers in creating resilient and sustainable food systems capable of adapting to future challenges.

Central to the FABLE approach is the FABLE Calculator, a powerful Excel-based tool designed to model and project sustainable pathways. The calculator includes 88 raw and processed indicators related to the agricultural sector, economy, and population [22]. A FABLE Pathway represents a combination of scenarios that outline the developments within a system along a specific trajectory. These scenarios are selected by the end-user—in this case, the authors as leaders of the Greece FABLE team—based on assumptions about key economic, institutional, climatic, and social variables. In turn, the scenarios encompass all of possible actions that determine the trajectory of the chosen pathway.

The FABLE Calculator utilizes a continuously updated dataset for all agri-food indicators drawing from the FAOSTAT database, which provides detailed data on agricultural production, yields, and harvested areas for various crops and livestock. This historical data is crucial for establishing baseline scenarios and calibrating projections according to the selected pathway. The baseline, or business-as-usual scenarios, depict an extrapolation of current practices over a 30-year horizon from 2020 onward. Additionally, user-implemented pathways modify the evolution of relevant variables according to specific scenarios, allowing for tailored shifts in productivity, land use, and other critical factors.

For example, in the case of crop and livestock productivity, baseline scenarios reflect productivity growth from 2000−2010, calculated using FAOSTAT production data. A high productivity growth scenario might involve adjustments such as multiplying historical growth rates by −1 if they were negative, by 2 if they were below 1%, and by 0.7 if they exceeded 1% [22]. These adjustments allow the model to project future productivity under varying assumptions.

The FABLE Calculator uses a sophisticated integrated modelling framework that combines data on (inter alia) agricultural productivity, land use, agricultural practices, economic activity, dietary patterns, to develop comprehensive scenario analysis, considering feedback loops to reflect how changes in one area (e.g., increased crop yields) can affect other areas (e.g., land use, biodiversity). The calculator provides outputs on a bevy of indicators pertaining to land-use and food systems, including detailed greenhouse gas emissions, biodiversity conservation, land and water use, land use efficiency, food production and food trade balance, and agricultural employment.

The process starts with the user inputs including assumptions for all 22 scenarios that constitute a pathway. As indicated in Fig 1, this process defines the targeted human consumption, which is the distillation of all the assumptions made by the user. This, in turn, mobilizes the dynamic simulations over the 2020–2050 period by calculating the necessary livestock and crop production to meet the targeted consumption (steps 2 and 3). This leads to computations of the required land allocation for crop and livestock production (step 4). In steps 5–8, the feasible production is calculated

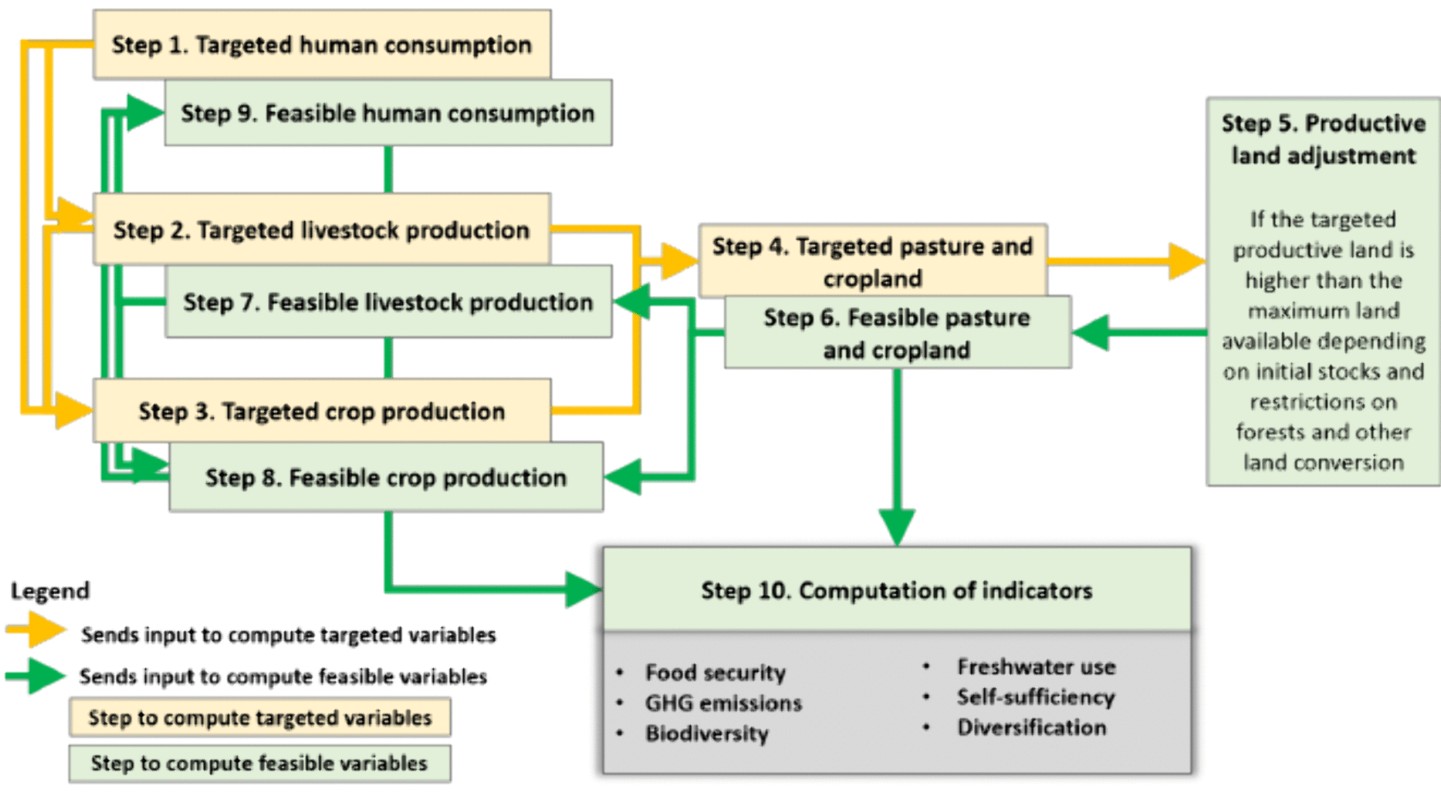

**Fig 1. FABLE Calculator.** Source: Mosnier et al. (2020) p. 23.

based on physical constraints, such as land availability and use changes (e.g., afforestation/deforestation), as determined by the scenarios in the pathway. Finally, the model shapes the feasible human consumption (step 9) which is the decisive factor behind all FABLE outputs described for the medium- and long- run. Section 3 describe the assumptions behind the baseline and national commitments pathways developed by the Greece FABLE team for the 2023 Scenathon. All negative productivity growth rates are multiplied by −1 in this scenario. The exception is milk from sheep and goats, whereby the 2000–2010 average of 1.42% is multiplied by 0.7 as per the FABLE Calculator operation. It is within this framework that we attempt an indicative assessment of the effects of a productivity increase in the crop and livestock sector in Greece.

## 3. Results from the FABLE Calculator

This section presents the indicative results from the scenarios described in Section 2 using the FABLE Calculator. Enhancements in crop and livestock productivity are included first in the BAU pathway and, as a second step, are integrated within a set of reforms derived from Greece's national commitments (see Table A1 in the S1 Appendix).

### 3.1. Business as usual and enhanced productivity pathways

**3.1.1. Construction of the pathways.** The first set of projections illustrates the change in key AFOLU variables and indicators from an exogenous increase in Crop and Livestock productivity whilst maintaining a Business as Usual (BAU) scenario for all other aspects, following the "Current Trends" pathway for Greece (see Appendix I). Using the 22 available levers in the FABLE Calculator, the baseline pathway assumes no significant policy and behavioural changes in Greece for the 2020–2030 horizon. Following trends of the 2000–2015 period, this pathway encompasses a minor

uptick in economic activity, no expansion in national protected areas and no change in national dietary patterns. Stagnant agricultural productivity is defined as the 2000–2010 average productivity growth for the livestock sector and unchanged productivity at the 2020 level for crop agriculture. Furthermore, the trajectory is calculated assuming no substantial shift in biofuel demand, no afforestation target, and no change in post-harvest losses. The baseline pathway representing current trends in the Greek AFOLU sector is embedded in a global GHG concentration trajectory that would lead to a radiative forcing level of 6 W/m2 (RCP 6.0), or a global mean warming increase likely 2–3°C above pre-industrial levels.

To illustrate the dynamics uncovered by an increase in agricultural productivity we proceed in a two-fold fashion. First, we introduce the boost in agricultural productivity within the BAU scenario, leaving all other policy parameters unchanged. In the case of the livestock sector this refers to an absolute reversal of the negative average productivity growth of the 2000–2010 for pigs, cattle, cattle milk, and eggs. All negative productivity growth rates are multiplied by −1 in this scenario. The exception is milk from sheep and goats, whereby the 2000–2010 average of 1.42% is multiplied by 0.7 as per the FABLE Calculator operation. High productivity growth in the crop sector is associated with a closure of the yield gap by at least 80%, compared with stability in the baseline scenario.

Having established the projected changes brought upon the Greek agricultural and food system following an increase in crop and livestock productivity in a stand-alone fashion, we turn to include the productivity shifts in the "National Commitments" pathway Greece (see Appendix II). The quantitative and qualitative targets which are considered for this pathway are extracted from a thorough review of key policy and legislative documents at the national and the EU level, such as Greece's NECP, the Development Plan for the Greek Economy [4]and all relevant documents from the EU Green Deal. Apart from increased productivity in the crop and livestock sector, the pathway is underpinned by medium to high speed of economic growth, shift to a healthy diet (as prescribed by the Lancet committee – EAT Lancet), a pronounced improvement in the country's agricultural trade balance reflecting the strategy for outward-oriented economic growth and productivity is expected to surge both for crops and for livestock production. This Pathway is embedded in a global GHG concentration trajectory that would lead to a lower radiative forcing level (RCP 4.5) and assumes expansion of protected areas and an increase in the deployment of organic practices in agricultural land. The detailed assumptions of both FABLE pathways are illustrated in Appendix I.

**3.1.2. Results.** Maintaining all policy and technology aspects in the baseline scenario except for an upward productivity shift in the crop and livestock sector significantly alters the agricultural GHG emissions pathway towards mid-century as shown in Fig 2. Total emissions are halved compared to the baseline scenario reaching 3 Mt in CO2 equivalents by 2050, thus recording a 29% reduction in comparison to 2020 levels. Furthermore, this underscores a cumulative 73.4% drop from the GHG emission levels for Greek agriculture in 2000, indicating sharp compliance to EU green transformation objectives.

The result is mainly driven from the drastic reduction in livestock emissions (2.47 Mt in 250 compared to 4.53 Mt under the baseline scenario) as well as the elevated emission withdrawals induced by land use change (3.28 Mt in 250 compared to 2.31 Mt under the baseline scenario). The latter reflects the reduced needs for land expansion as the enhanced productivity increases yield without requiring vast amounts of input, corroborated by the decline in pastureland (Fig 3). Agricultural emissions tied to crop production also diminish under the high productivity scenario, diverging significantly from the baseline after 2035 according to the projections. Under the assumption of high productivity growth, nitrous oxide and methane emissions drop significantly compared to the current trends scenario, marking a reduction of 11% by mid-century for both greenhouse gasses. The downward trend is more pronounced for methane, whereby emissions fall by more than 18% in the 2025–2050 interval, largely driven by the 30% drop in livestock emissions. The relevant drop in nitrous oxide emissions is almost 5.6% for the same period. However, it is noteworthy that this is driven by the 29% drop in livestock emissions which offset the 6% increase from the crop sector emissions (which are higher in magnitude throughout the 25-year period). The latter could reflect increased fertilizer use and more intensive manure application embedded in the efforts to enhance productivity.

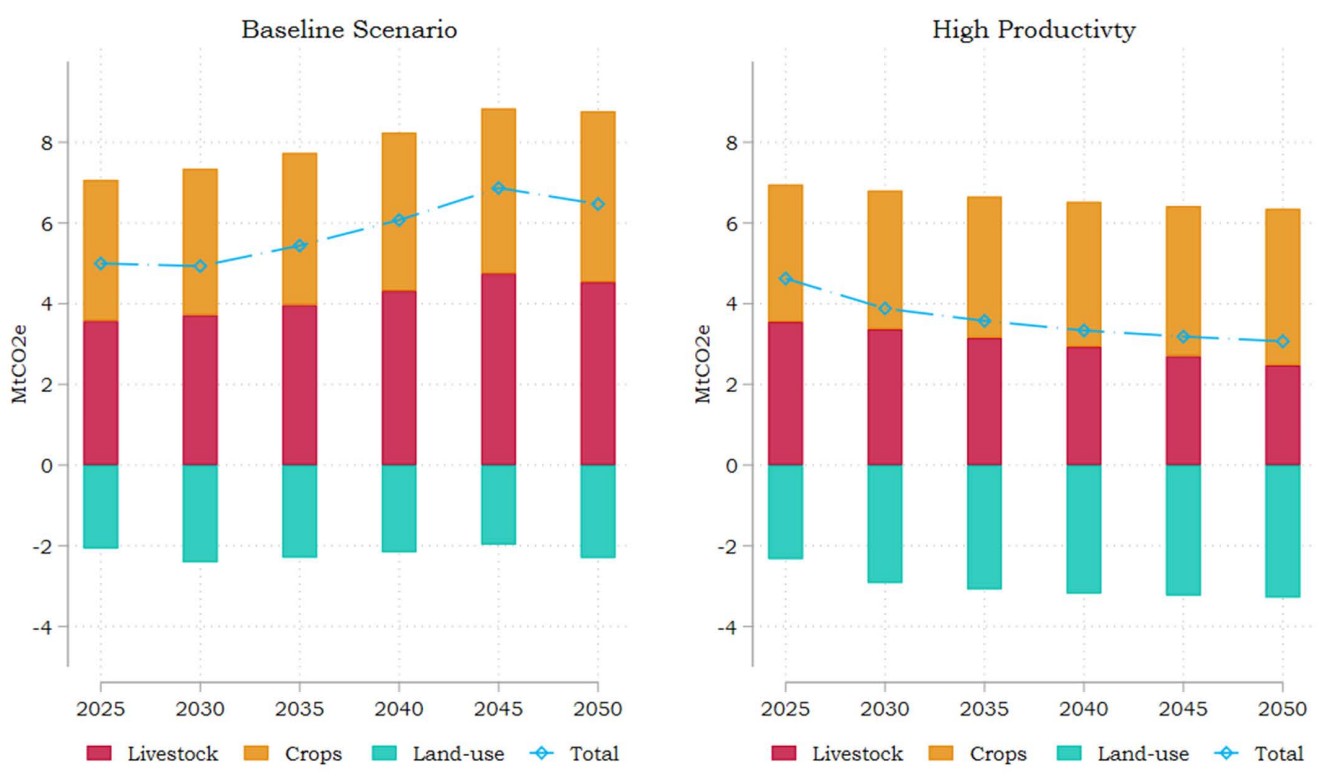

**Source: FAO and Authors' Calculations**

**Fig 2. AFOLU Emissions – Baseline and Increased Productivity Pathways.** Source: Authors' calculations.

It is documented in the literature that advances in productivity allow farmers to produce more food on the same amount of land, thereby reducing the pressure to expand into forests and other natural habitats [23]. Another study underlines that enhanced crop yield growth have saved an estimated 290 million hectares of cropland and avoided the expansion of 120 million hectares of grassland by 2030 [24]. Furthermore, simulations using the GLOBIOM model indicate that closing yield gaps by 50% for crops and 25% for livestock by 2050 could decrease emissions from agriculture and land-use change by 8% overall [25]. Khatri-Chhetri et al find evidence of technical efficiency improvements reducing agricultural emission intensity in India, as production inputs are used more efficiently to close yield gaps [26]. Precision agriculture technologies utilize sensors and data analytics to match fertilizer application to crop needs. This reduces nitrogen surplus, which is a primary driver of nitrous oxide emissions [27]. Northrup et al. find that such digital tools can reduce nitrogen fertilizer application by 36%, leading to a 23% reduction in emissions [28].

As a result of the substantial increase in crop and livestock productivity, the area dedicated to pastureland drops by more than 55% over the 2025−50 period, compared to a 22% drop in the Current Trends pathway (Fig 3). In the FABLE framework, higher productivity implies that the same level of animal production can be achieved with fewer animals, which directly reduces the demand for grazing land. Since pastureland requirements are proportional to livestock numbers and stocking densities, productivity improvements translate into non-negligible reductions in pasture area, even when total consumption patterns remain unchanged. In FABLE, pastureland is governed by minimum and average stocking density constraints. When livestock productivity improves, fewer livestock units are required, pushing effective stocking rates

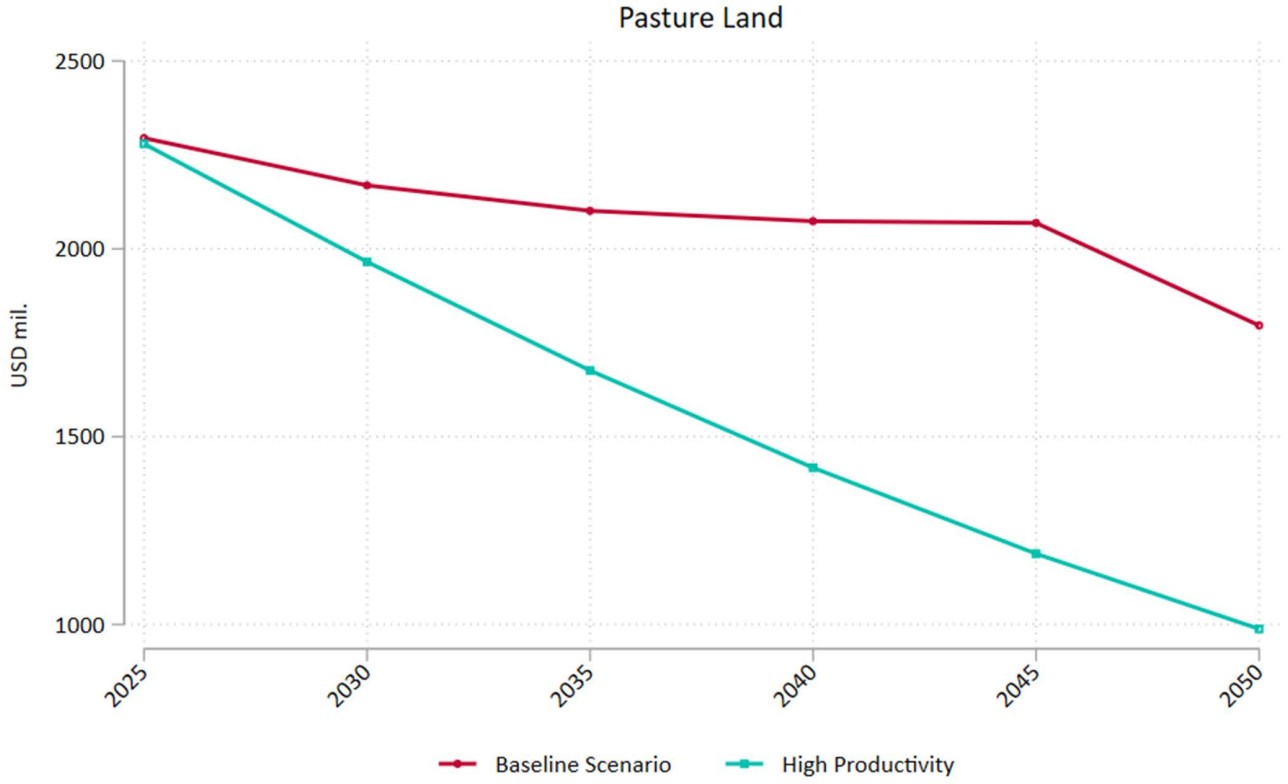

**Fig 3. Pastureland – Baseline and Increased Productivity Pathways.** Source: Authors' calculations.

toward the lower bound. Once stocking densities approach minimum thresholds, pastureland becomes structurally redundant and exits production, leading to accelerated land contraction. This creates a threshold effect, explaining why pastureland falls much faster under High Productivity than under Current Trends.

An important aspect of a surge in agricultural productivity is the double dividend of climate change mitigation and increased competitiveness of the domestic agricultural sector. Fig 4 clearly illustrates the diverging trajectories in production costs for Greek agricultural producers, a heavily debated issue following the 2023 extreme weather events and the 2023–27 CAP. Total costs gradually decrease under the assumption of high productivity from 828 million euros in 2025 to less than 630 million in 2050, driven predominantly by diminishing pesticides costs, which represent the lion's share in total expenditures. Specifically, the producers' expenses on fertilisers mark a 27.5% reduction in the 2025–2050 period in the high productivity scenario thus representing just the two fifths of the respective cost under the business-as-usual scenario. Fertilizer costs drop somewhat less impressively by 14.8% over the 25-year period, however the divergent upward trajectory in the baseline scenario results in a cost saving gain of almost 40% for domestic producers because of enhanced productivity in the sector. Machinery costs also drop by more than a fifth in the high productivity pathway compared to a commensurate increase under current trends, most probably reflecting the non-negligible reduction in pastureland. The detailed assumptions of both FABLE pathways is illustrated in Appendix I.

Our results regarding agricultural production and costs are also in line with the findings in the literature. Devot et al. study the impact of extreme climate events on agricultural production in the EU to showcase how the increasing frequency of extreme weather events, exacerbated by climate change, poses significant challenges for agricultural productivity and

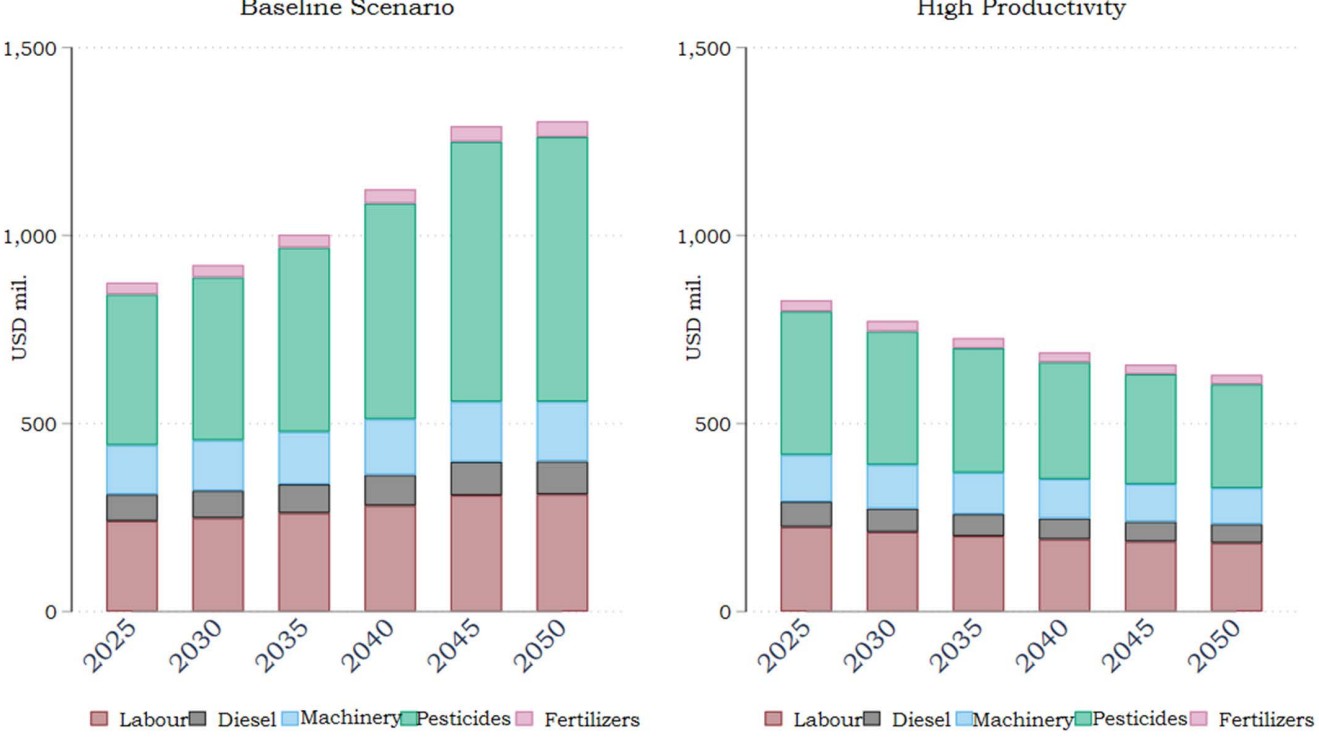

**Fig 4. Production Costs – Baseline and Increased Productivity Pathways.** Source: Authors' calculations.

costs [29]. The efficient use of inputs, from fertilizers to machinery, is highlighted by [30] as a critical force for reducing production costs following new technology adoption. For the case of Greece, acknowledging the high degree of land fragmentation, productivity increases can bring forward economies of scale which, in turn, contribute to non-negligible reductions in production costs [31].

### 3.2. Enhanced productivity in a national commitments pathway

As a further exercise, we incorporate the productivity surge in the wider range of reforms pertaining to the implementation of all national commitments outlined in this Section (see Table A2 in S1 Appendix). As shown in Fig 5, a shift to higher agricultural productivity coupled with behavioural change, pronounced economic growth and more stringent monitoring of environmental policies results in a substantial abatement of GHG emissions from agriculture in Greece, reaching 1.6 Mt CO2e in 2050. The latter represents a 62% drop from 2020 levels in 30 years, a 74.4% reduction compared to baseline 2050 emissions and a 46% reduction compared with the High Productivity scenario. The emission abatement is mainly driven by the pronounced 46% reduction in methane ($CH_4$) emissions, which is primarily driven by structural changes in the livestock system and consumption patterns, rather than by incremental efficiency improvements alone. The decline in ruminant meat consumption under the healthy diet scenario included in the pathway is the main driver behind this, given that enteric fermentation in ruminants constitutes the dominant source of agricultural methane. This effect is further reinforced by higher livestock productivity, which reduces emissions per unit of output,

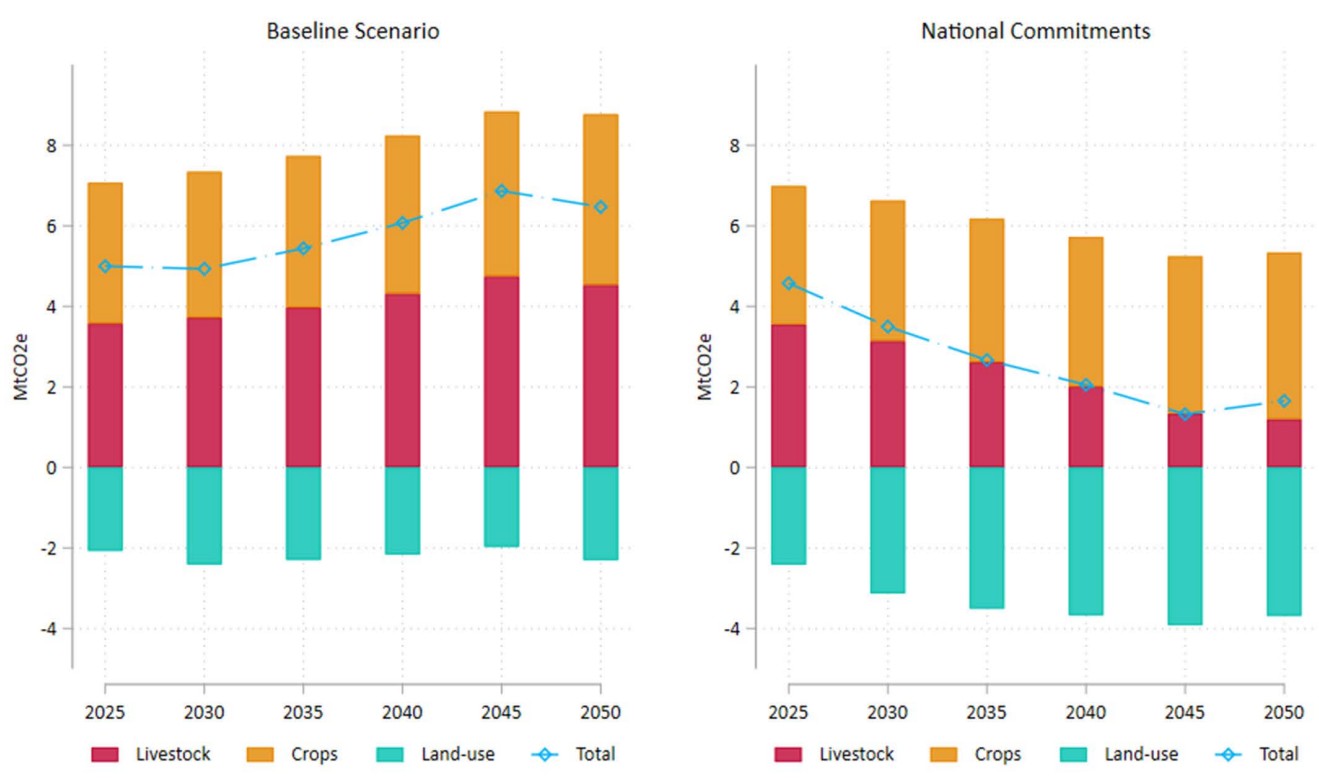

Source: FAO and Authors' Calculations

**Fig 5. AFOLU Emissions – Baseline and National Commitment Pathways.** Source: Authors' calculations.

and by the sharp decline in pastureland, limiting extensive grazing systems that are methane intensive. This sizeable drop in CH4 emissions contributes to the overall emission-reducing effect of this pathway despite the modest increase if nitrous oxide emissions by 15% over the same period. This most probably reflects the higher and more stable crop output required to meet domestic consumption targets as plant-based food demand increases. These results clearly indicate the need for a holistic approach to the sustainable transformation of the agricultural sector, whereby support for technological advances in the crop and livestock sector are commingled with behavioural change, environmental policy stringency and sound economic policies.

Livestock emissions drop by two thirds over the 2025−50 period and drive the overall emissions abatement by large in this scenario, more than the High Productivity case (a 33% reduction by 2050 when shifting from high productivity to national commitments pathways) since meat consumption is expected to fall as the population shifts to healthy dietary patterns. Both nitrous oxide and methane emissions decrease by approximately this amount over the 25 year period Emissions related to crop agriculture increase by 20% by mid-century and are marginally elevated (6.5% in 2050) compared to the High Productivity Pathway, reflecting the change in demand towards plant-based dietary patterns, however, remain moderately subdued compared to the baseline scenario. Finally, emissions savings from land use and land use change are modestly increased under the National Commitments scenario (a 52.7% drop by 2050 compared to 41% under the High Productivity pathway), reflecting the clearly defined afforestation targets and the halt of urban area expansion assumed.

Fig 6 shows the marked decline in pastureland compared to the BAU pathway when we complement productivity increases with a broader set of reforms compatible with Greece's national commitments. In line with the high productivity pathway depicted in Fig 2, we observe a sharp drop in terms of hectares devoted to pastureland, however the levels of 2050 in the national commitments pathway represent only a quarter of the ones implied by BAU. Abiding to all official commitments for the Greek agri-food sector results into a sizeable drop in pastureland to less than 500 thousand hectares by 2050, representing an 80% reduction compared to 2025 levels. Furthermore, total pastureland is less than half of that calculated if we just assume high productivity within a baseline framework. Applying the high productivity scenario leads to a 57% decrease in pastureland with respect to 2025, which is also impressive albeit considerably smaller than the national commitments pathway. The result is primarily driven by the assumption of a shift to healthy dietary patterns (following the LANCET recommendations) in the national commitments pathway, underscoring the synergies between supply and demand factors in the sustainable transition of the agri-food sector in Greece. It must be noted that all individual GHG categories record a marked drop in emissions under the national commitments pathway, including nitrogen oxide emissions from crop agriculture. The latter, which increased under the high productivity scenario, drop by 11.2% by 2050 indicating that the concomitant shift in demand (mostly attributed to changing dietary patterns) offsets the need to intensify fertilizer and manure use to boost productivity.

The results are in line with the relevant literature, as diet-based reforms appear to yield significant emission mitigating effects for the AFOLU sector [32,33]. In the United Kingdom, sustainable pathways that increase productivity alongside dietary changes could free up land for nature recovery, potentially turning the sector into a net carbon sink by 2040 [34]. Aggregate results from all FABLE countries for 2023 document the substantial decrease in GHG emissions from agriculture under the *National Commitments* Pathway compared to current trends [35]. Another study corroborates that shifting to healthy diets by far enhances the GHG mitigating effect of productivity increases, with their GLOBIOM projections pointing towards a 25–27% drop in AFOLU emissions by 2025 in the US under the healthy diet scenario [36].

In addition, to corroborate our results, meta-analyses and IPCC synthesis reports document that $N_2O$ emissions rise with increasing nitrogen inputs and can exhibit nonlinear responses as fertilizer rates exceed crop needs, which helps

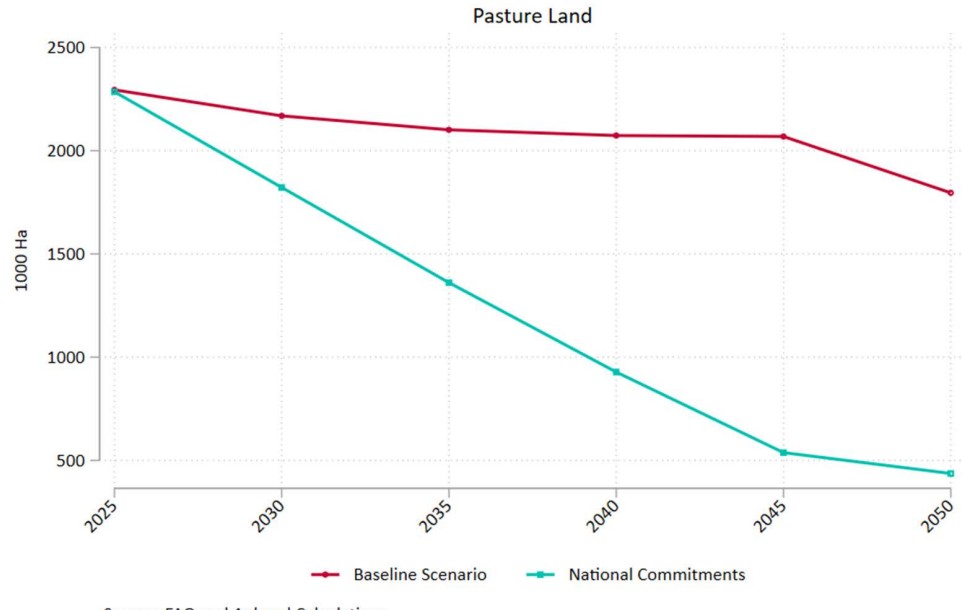

**Fig 6. Pastureland – Baseline and National Commitments Pathways.** Source: Authors' calculations.

explain why some "more sustainable" pathways may still face trade-offs in soil $N_2O$ unless complemented by precision nutrient management [37,38].

The implications for agricultural production costs shown in Fig 7 are fairly similar to the High Productivity baseline scenario (4). Incorporating the assumptions for elevated crop and livestock productivity in a pathway of national commitments reduces production costs by 700 million in 2050, thus a further curtailment of costs by 25 million compared to the high productivity pathway (a 4% decrease between the two). The sharply reduced use of pesticides explains the bulk of the result in this case as well. Hence, the main factor behind cost saving is the increase in productivity in the crop and livestock sector, which can be fostered through the adoption of cutting-edge technologies. The literature provides credit to the view that a holistic reform package including supply and demand measures, in line with the ones underpinning the national commitments pathway in the FABLE Calculator, can drastically ameliorate production costs [39]. Dietary shifts toward healthier and more plant-based diets reduce GHG mitigation costs substantially when combined with supply-side agricultural improvements [40]. They show that with both demand changes and supply efficiency gains, the total cost of mitigation in the food system decreases relative to scenarios with rigid diets. Havlik et al. use the GLOBIOM model to show that dietary shifts combined with yield improvements and reduced deforestation deliver the largest cost-efficient mitigation (including reductions in agricultural emissions) compared with isolated actions [41].

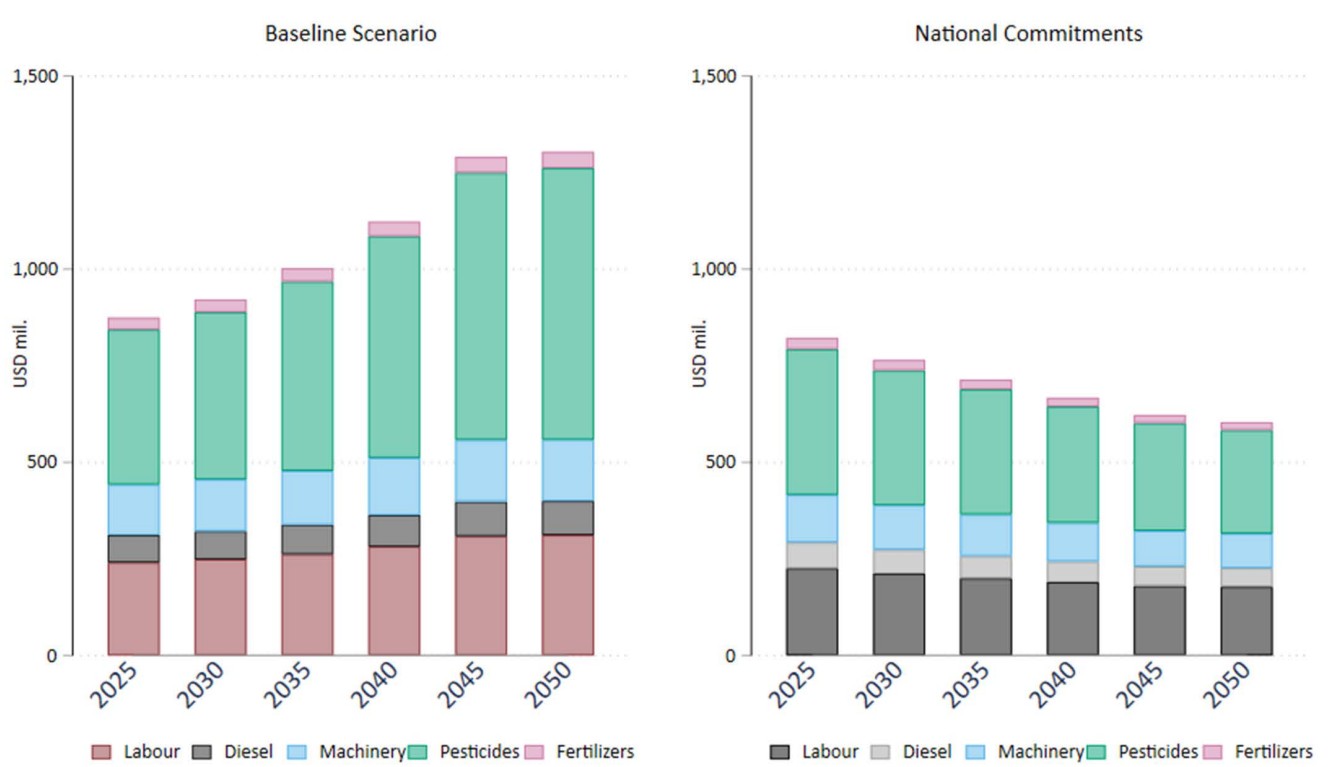

**Fig 7. Production Costs – Baseline and National Commitments Pathways.** Source: Authors' calculations.

Figs 2–7 clearly depict the double dividend to be made from bolstering agricultural productivity in terms of minimizing costs for the producers and promoting emission reduction targets with multiple gains for society. While the reigning in of costs is heavily dependent on productivity, commingling supply side with demand side policies ushers in significant gains in the environmental front.

Fostering productivity growth is a key policy target for the Greek agricultural sector. Having said that, employment in the sector remains above 10% and places Greece among the top three spots in the EU's relevant statistics. Hence, supporting agricultural jobs is pivotal to the overall national economic development strategy [4,42]. As shown in Fig 8, promoting agricultural productivity growth yields benefits for agricultural employment, however much more so when it is embedded in an integrated reform framework towards sustainable agriculture.

Employment gains in the National Commitments scenario materialise from 2035 onwards and result in a steady increase in active jobs throughout the 30-year period. By 2050 the tangible effect is 24 thousand more employment positions (FTE) compared to the baseline scenario and 13 thousand against the stand-alone productivity enhancement pathway for Greece. The result is driven by the uptick in economic activity assumed in the National Commitments and even though no growth in irrigated harvested area is assumed, highlighting the decoupling potential of inclusive economic growth and GHG emission reduction.

## 4. Policy recommendations for enhancing agricultural productivity

Section 3 presents the large, untapped potential for the Greek agri-food system under a rational productivity increase in the crop and livestock sector. The reduction in GHG emissions coupled with a substantial drop in production costs is impressive even when keeping all other policy levers unchanged, as per the current trends pathway. Moreover, applying the ambitious set of policy reforms under the national commitments pathway enhances these results and marks a substantial improvement of Greek agriculture towards sustainability (reflected in reduced emissions) and competitiveness (through the drop in production costs). It is, therefore, imperative that the policy mix target the underlying drivers for

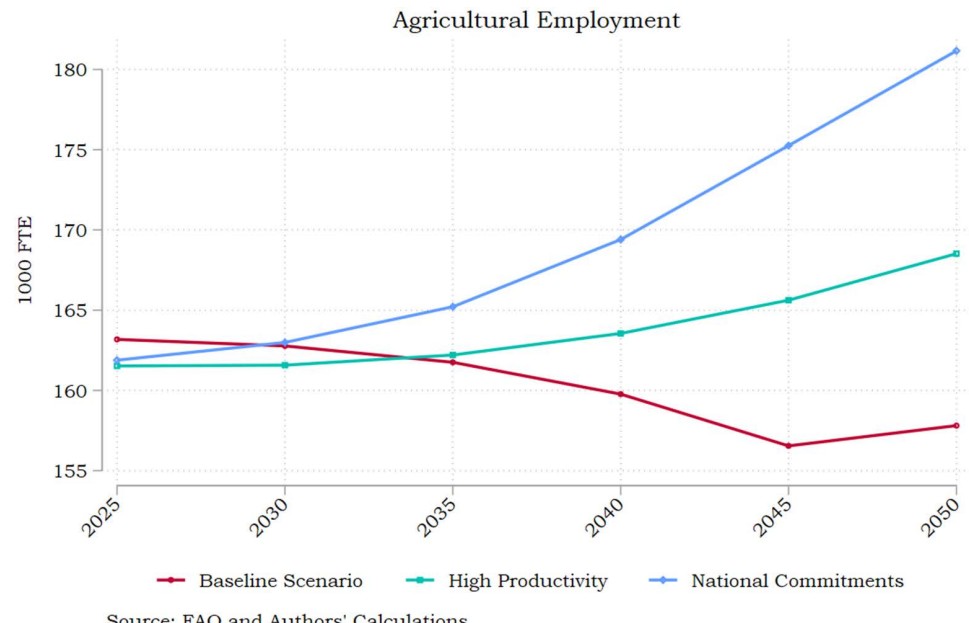

**Fig 8. Agricultural Employment – All Scenarios.** Source: Authors' calculations.

boosting agricultural productivity growth. To unlock latent productivity dynamics, decision makers need to focus on (i) the functioning of national and sub-national innovation systems to produce tangible results for the agri-food sector, (ii) life-long education and learning, upskilling and reskilling of the agricultural workforce and (iii) reforms to increase the size of agricultural holdings.

Recent research emphasizes that sustainability transitions require integrated innovation systems rather than isolated technological fixes. Holistic and sustainable transitions emerge from coordinated advances in technology, policy, finance, and institutional design [43]. This perspective underpins the horizontal and vertical policy measures proposed here, which aim to align productivity-enhancing innovations with environmental and socio-economic objectives

### 4.1. Horizontal policies

In essence, promoting innovation and technological change in the Greek agricultural sector is an integral part of the overall technological transformation of the Greek economy. According to the EU Innovation Scoreboard, Greece is classified as a Moderate Innovator within the EU, showing strong dynamics in tertiary education and innovative SMEs [44]. However, the country is traditionally lagging in building a domestic innovation system (and actively participating in regional innovation systems), which would ensure the dissemination of knowledge and foster inter-sector collaborations. The lack of integration of the business and -even more- the agricultural sector in the knowledge creation and dissemination process has been a standing characteristic of the Greek post-war economy [45]. The Greek economy has struggled to commercialize R&D efforts into patents, trademarks, and design applications. More critically for agriculture, there is a lack of strong inter-sectoral ties that would ensure a seamless pipeline from research to production [46]. Promoting linkages between knowledge creation and application has been a long-time challenge for technological transformation in the Greek economy and tackling it could yield significant dividends in enhancing well-being and meeting sustainability targets [47].

Enhancing lifelong learning and expanding it to actively cater to the agricultural population's needs is pivotal for fostering technological transformation in the agricultural sector. The learning process needs to actively connect the research and innovation sector to the application of innovative practices, highlighting the mitigating effects on GHG emissions as well as the employment- and growth enhancing effects of key technologies that boost agricultural productivity. Since skills play a crucial role in harnessing the productivity gains associated with precision agriculture and advanced technology, incorporating in-the-field activities into the academic curricula of Greek universities related to agriculture will bolster the interlinkages between knowledge and practice. Under enhanced productivity, agricultural GHG emissions decline by 29% by 2050 in the BAU case and by 62% under national commitments, while total production costs fall by nearly 50%, driven primarily by pesticide and fertilizer reductions. These outcomes can only be realized in practice if farmers possess the technical capacity to implement input-efficient technologies.

Bold reforms in the way the country perceives learning can thus unlock significant potential for enhancing crop and livestock productivity. Finally, increasing the average size of Greek agricultural holdings through incentives for cooperation and clearly defined property rights is a crucial step towards leveraging economies of scale and boosting productivity [4].

### 4.2. Vertical policies

To further encourage the adoption of precision agriculture technologies in Greece, a series of vertical policies and initiatives can be implemented. These efforts aim to provide comprehensive support to farmers, develop sustainable technological infrastructures, and address the unique geographical challenges of the country.

A pivotal framework in this effort is the Agricultural Knowledge and Innovation Systems (AKIS), which connects farmers, research institutions, and the private sector. By facilitating knowledge transfer and collaborative innovation, AKIS helps overcome barriers such as high initial costs and technical complexity, providing tailored training and support. Supported by the CAP 2023–2027, AKIS funds initiatives that promote sustainable and competitive agricultural practices, ensuring a holistic approach to modernizing Greek agriculture [48]. The dominant contribution of livestock emission reductions to total

agricultural mitigation efforts under both high productivity and national commitments pathways underscores the importance of AKIS interventions targeted specifically at precision livestock management, animal health monitoring, and feed optimization

Additionally, creating a national registry of technical support for precision agriculture technologies can enhance the adoption of new practices among farmers. This registry would include precision agriculture experts who offer tailored advice and recommendations based on each field's specific characteristics. An example of this is the Farm Advisory System (FAS) by the Greek Ministry of Rural Development and Food, which provides advisory services to improve agricultural practices and technology use.

Establishing a national registry of local sensor systems, combined with digital infrastructure for processing open data, can effectively address Greece's diverse geomorphology. This initiative involves setting up local sensor networks and a digital framework to analyze open data, enabling better management of agricultural landscapes. For instance, the GAIA Sense project in Greece uses sensors to collect real-time data on soil conditions, weather, and crop health, providing actionable insights for irrigation, fertilization, and pest management [49]. Such technologies optimize resource use and enhance productivity. Smaller, localized projects can ensure that all farmers, regardless of scale, have access to advanced tools. The increase in crop-related nitrous oxide emissions under the high-productivity pathway, followed by an 11.2% reduction under national commitments, highlights that productivity alone is insufficient to control fertilizer-driven emissions. This directly motivates the development of national sensor registries and precision nutrient management systems, which are structurally required to decouple yield growth from nitrogen surplus.

Moreover, regional digital hubs focusing on precision agriculture can serve as centers of excellence, offering access to the latest technologies and training. These hubs can host workshops, demonstrations, and pilot projects to showcase the benefits of precision agriculture. The "Smart Farming Initiative" exemplifies this approach, providing farmers with access to advanced technologies and support services, promoting widespread adoption of precision farming practices.

To support these advancements, innovative financial models, such as micro-financing options and low-interest loans tailored for precision agriculture, can be introduced. These financial instruments can help small farmers afford the initial costs of adopting new technologies, making advanced tools more accessible to a broader range of farmers [3]. Following the projections in Section 3, one can postulate that the large pesticide and fertilizer cost savings justify targeted microfinance and concessional loan schemes to overcome initial adoption barriers and unlock the cost-reduction trajectories under the high productivity and national commitments pathways.

## 5. Discussion

The results derived from the FABLE Calculator provide optimistic scenarios and strong incentives for Greece to tackle its most critical agricultural challenges. The application of PA technologies aligns seamlessly with these results, capturing the three dimensions of sustainable production: environmental, economic, and social, which are the primary outcomes highlighted by the FABLE Calculator.

Regarding our first research question, the FABLE Calculator projects a significant reduction in agricultural emissions through enhanced livestock and crop productivity, with reductions ranging from 29% to 62% by 2050, depending on the selected FABLE pathways. These results are primarily driven by the decrease in pastureland highlighting the synergistic effect of supply and demand factors in the national commitments scenario. Additionally, enhanced productivity contributes to significant cost reduction for producers, while maintaining or even increasing agricultural employment in Greece.

Our findings corroborate with numerous studies in this field. Results from the FABLE show that productivity increases are key in curtailing agricultural emissions and meeting national and global sustainability targets, particularly when combined with demand-side effects from healthier dietary shifts [50]. Similarly, results from the GLOBIOM model show that closing yield gaps by 50% could lead to a significant decrease in global GHG emissions in the coming decades [25].

Studies also report robust but modest emission reduction in the US agricultural sector due to higher yields, emphasizing the potent impact of healthier dietary patterns [36].

Considering our second research question, we determine that PA technologies offer the most multifaceted positive impacts on Greece's agricultural sector, which is characterized by several unique traits. The sector is predominantly composed of small to medium-sized, often family-owned farms, with an average farm size significantly smaller than the European Union average [51]. The workforce is aging, with the average age of farmers exceeding 55 years [52]. Moreover, the sector faces significant environmental challenges, including soil degradation due to continuous land use for agricultural purposes [53]. Furthermore, water consumption for agriculture accounts for 74% of the total, indicating that agriculture is the largest water consumer in Greece [54].

Underlying research concludes that PA technologies can address a wide range of these challenges. For example, studies found that adopting PA practices in Greek dairy farms significantly improved sustainability and reduced greenhouse gas emissions [55]. The analysis of data from 240 farmers, reveal that technologies such as AI and IoT, in addition to precision agriculture, enhance resource efficiency, reduce environmental impacts, and increase yields [20]. Additionally, research on precision irrigation technologies in olive and cotton farming in Messinia and Thessaly highlights potential benefits such as optimized water use and increased crop yields [18]. These findings underscore the significant advantages of adopting PA technologies, and the productivity gains demonstrated by the FABLE Calculator.

Other technologies, such as organic farming, agroforestry, and GMOs, also show potential for increasing productivity. However, further research is necessary to fully understand their applicability and benefits in the Greek context.

Pertaining to our third research question, we emphasize the potent role of horizontal and vertical policies in promoting the development, adoption and dissemination of cutting-edge technologies such as precision agriculture. The Greek agricultural sector faces significant challenges in advanced technologies due to its low capacity, a lack of functioning innovation systems, and a workforce with limited skills. Given the considerable externalities associated with new technologies and the low technological capacity of the Greek agricultural sectors, targeted policy interventions are essential. Horizontal policies should focus on boosting lifelong learning to enhance the skills of the agricultural workforce, and incentivizing research and development (R&D) in the agricultural sector through fiscal incentives and frameworks for tangible collaboration with the research community. Vertical policies include strengthening the Agricultural Knowledge and Innovation Systems (AKIS) and establishing a national registry of technical support for precision agriculture technologies. This way farmers gain farmers access to the most updated resources, information and tools for applying cutting-edge technologies in the field.

Our suggestions are in line with the policy propositions of [4], who emphasize the importance of fostering ties with the research community and developing digital and technological skills as key policy interventions to transform the Greek agricultural sector. The literature also recognizes the productivity growth potential of precision agriculture and the urgent need for integrated policy measures to foster it [5]. His proposed two-pronged approach combines macro-level interventions to bolster infrastructure with micro-level policies to map farmers' capabilities, enhance skills, and promote new technologies. Scholars highlight the importance of PA technologies at the European level, arguing that bold policy interventions are required to promote their adoption [30]. They note that the EU agricultural sector will benefit from tailored data solutions, such as free provision of geospatial mapping data at fine resolutions and a surge in investment for enhancing learning and skills for the workforce.

Commingling cross-cutting and bespoke policies for technology adoption in agriculture in material. Agricultural policy does not only shape incentives but also farmers' attitudes toward risk and uncertainty. Changes in EU agricultural policy significantly affect farmers' risk preferences, with direct implications for investment decisions and technology adoption [56]. This behavioural channel helps explain why productivity-enhancing technologies may diffuse slowly even when they are economically viable, constituting policy support indispensable.

 

## 6. Conclusion

Sustainable food and agricultural systems are paramount for Greece's climate-resilient development, especially given the nation's geographical and climatic vulnerabilities. Greece is increasingly facing challenges such as water scarcity, soil degradation, and extreme weather events exacerbated by climate change. Addressing these challenges requires a paradigm shift towards sustainable agricultural practices and food systems. This transition hinges on the adoption of technological advances and innovative solutions that enhance crop and livestock productivity.

Our analysis, utilizing projections from the FABLE Calculator, demonstrated the significant potential of increasing crop and livestock productivity in meeting sustainability goals. In a business-as-usual scenario, following current trends in Greek agriculture and policy, productivity improvements could lead to a 29% reduction in GHG emissions from agriculture over the 2020–2050 period. This reduction is primarily driven by diminishing livestock emissions and elevated negative emissions from land use. When these productivity gains are combined with shifts aligned with Greece's national commitments, the total reduction in agricultural emissions could reach 62% over the same period. Moreover, these scenarios significantly reduce the cost burden on producers, largely due to the sharp decrease in pesticide use in agricultural production. Supporting agricultural productivity growth, alongside a surge in economic activity, shifts to healthier dietary patterns, and more stringent environmental policies, can greatly contribute to emission abatement, job creation, and increased competitiveness in the Greek agricultural sector over the next 30 years.

The strengthening of agricultural productivity in Greece is intrinsically tied to the adoption of advanced technologies, particularly PA. The sector, facing numerous challenges—exacerbated by external factors such as climate change and economic pressures—requires innovative solutions to sustain growth and competitiveness. PA technologies stand out as a critical solution, offering precise, data-driven methods to optimize resource use, enhance crop yields, and reduce environmental impact, achieving significant gains in sustainability, including reduced greenhouse gas emissions and improved soil health. By adopting these technologies, Greece can build a more sustainable and resilient agricultural system, better equipped to meet both current and future demands.

Horizontal measures, such as enhancing digital skills within the agricultural sector and reforming formal and lifelong learning to facilitate the dissemination of knowledge into practical applications, are essential. Additionally, vertical policies, including the establishment of a national registry for precision agriculture, the development of local sensor networks, and the creation of regional digital hubs, are vital for facilitating technology adoption, enhancing accessibility, and driving sustainable growth.

Productivity enhancement is one of the key tenets of the transformation of the Greek agricultural sector. Its benefits extend beyond economic and efficiency gains, actively catalyzing emission reductions as well. Unlocking high productivity potential requires the development and adoption of cutting-edge technologies, particularly precision agriculture. This necessitates bold policy initiatives and widespread support to leverage innovation and enhance collaboration, enabling Greece to transition towards a more resilient and environmentally responsible agricultural sector while contributing to global climate objectives.

This paper uses the FABLE Calculator as the methodological tool for developing data driven projections for land use, emissions and production costs under different policy scenarios. The Calculator has limitations, as do all modelling tools. We are obliged to use the inherent definition of enhanced productivity (as described in Section 3.1) both for crops and livestock. We aim to add flexibility to the productivity growth scenarios in future research. Moreover, there is no spatial granularity within the country, meaning we apply the changes in productivity (and other policy levers) in a uniform fashion for the entirety of the Greek agricultural sector. Finally, we do not explicitly model the effect of key technologies such as precision agriculture on the variables of interest, as this is not available with the current version of FABLE Calculator. Nonetheless, the link between agricultural innovation and productivity growth is robust.

Ultimately, the sustained increase in crop and livestock productivity forms the backbone of a sustainable agri-food system in Greece. It ensures food security, enhances economic viability for farmers, and protects the environment, creating a balanced approach that supports long-term sustainability.

 

## Supporting information

**S1 Data. Data Projections.** A spreadsheet with the time series (FABLE-C Projections) for all the variables used in the manuscript.
(XLSX)

**S1 Appendix. Greece Current Trends and National Commitments.** Tables A1 and A2 present the description of the distinct pathways for the Greek agri-food sector, as documented in the FABLE 2023 Scenathon.
(DOCX)

## Author contributions

**Conceptualization:** Phoebe Koundouri, Konstantinos Dellis.

**Formal analysis:** Konstantinos Dellis.

**Methodology:** Olympia Miziaki.

**Supervision:** Phoebe Koundouri.

**Validation:** Phoebe Koundouri.

**Visualization:** Konstantinos Dellis.

**Writing – original draft:** Konstantinos Dellis, Olympia Miziaki.

**Writing – review & editing:** Konstantinos Dellis, Olympia Miziaki.

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
