## [Decision Letter · Decision Letter 0]

11 Nov 2025

Dear Dr. Dellis,

Thank you for submitting your manuscript to PLOS ONE. After careful consideration, we feel that it has merit but does not fully meet PLOS ONE’s publication criteria as it currently stands. Therefore, we invite you to submit a revised version of the manuscript that addresses the points raised during the review process.

Thank you for your careful and well-written manuscript. Both reviewers find your paper interesting and well aligned with PLOS ONE’s scope, acknowledging the robustness of the analytical framework and the policy relevance of the results.

Please revise the paper according to the reviewers’ specific points - particularly:

expand the introduction to include more contextual references on technological innovation and sustainability in Greek agriculture;clearly describe the dataset used in the FABLE calculator and ensure consistency between numerical values in the abstract and main text;correct minor stylistic and typographical issues, and reorganize Section 3.1 for better readability.These refinements will further enhance the clarity and impact of your work.

We look forward to receiving your revised manuscript.

Kind regards,

Federico Zilia

Academic Editor

PLOS ONE

Journal Requirements:

2. In the online submission form you indicate that your data is not available for proprietary reasons and have provided a contact point for accessing this data. Please note that your current contact point is a co-author on this manuscript. According to our Data Policy, the contact point must not be an author on the manuscript and must be an institutional contact, ideally not an individual. Please revise your data statement to a non-author institutional point of contact, such as a data access or ethics committee, and send this to us via return email. Please also include contact information for the third party organization, and please include the full citation of where the data can be found.

3. Please amend the manuscript submission data (via Edit Submission) to include author P. Koundouri and O. Miziaki.

4. We note you have included a table to which you do not refer in the text of your manuscript. Please ensure that you refer to Table 1

Reviewers' comments:

Reviewer's Responses to Questions

**Comments to the Author**

1. Is the manuscript technically sound, and do the data support the conclusions?

Reviewer #1: Yes

Reviewer #2: Yes

2. Has the statistical analysis been performed appropriately and rigorously?

Reviewer #1: Yes

Reviewer #2: I Don't Know

3. Have the authors made all data underlying the findings in their manuscript fully available?

Reviewer #1: Yes

Reviewer #2: No

4. Is the manuscript presented in an intelligible fashion and written in standard English?

Reviewer #1: Yes

Reviewer #2: Yes

Reviewer #1: Using the analytical power of the FABLE calculator to quantify the impact of improved crop and livestock productivity on key agricultural, forestry and Other Land use (AFOLU) environmental indicators, this article explores the transformative potential of the Greek agricultural sector to enhance crop and livestock productivity. The whole article is clear, but the following changes still need to be made:

1. It is suggested that the author include in the introduction the relevant research of other scholars on the combination of technological innovation and sustainability in the Greek agricultural sector, and summarize and analyze the innovation and advantages of this study compared with it.

2. In section 3.1, it is suggested that the author divide the relevant description of the productivity improvement path and the relevant results of the productivity improvement path into two parts for discussion, such as adding corresponding subheadings respectively, so as to make the content of this part more clear.

3. In Section 3.2, the author compares and analyzes the results of national commitment scenario and BAU scenario in AFOLU emission and pastureland, but lacks quantitative analysis of relevant results.

4. In the third part, the author discussed the differences in results under different paths, but the reasons for this lack of more in-depth demonstration, such as the support of relevant literature. It is suggested that the author supplement this.

5. When the author discusses horizontal policy and vertical policy in the fourth part, he should start from the perspective of the results of various indicators under the various paths of this paper, and there is no correlation between the two.

6. It is suggested that the author add the relevant discussion on the shortcomings of this study and the research issues that can be further explored in the future in the summary of Section 6.

Reviewer #2: The authors use the FABLE Calculator to assess the intricate relationship between improved productivity in crops and livestock on environment, showing that enhancing productivity reduced GHG emissions by up to 29% until 2030 and 62% by 2050. The authors compare BAU and Enhanced Productivity Pathways, taking into account policy documents. The authors provide evidence for the use of precision agriculture for increasing agricultural productivity in Greece. Moreover, horizontal and vertical policies for the development, adoption and dissemination of latest technologies are proposed.

General comments:

The paper elaborates on the use of technology-driven productivity surge in crop and livestock in Greece, using the FABLE calculator. However, the dataset summary is missing in the text. The reader therefore cannot make sense of what data was used for the analysis in the FABLE calculator.

Specific comments:

In the Abstract, the figures for reduction of GHG emissions are 21% until 2030 and 52 % until 2050. However, in the text, the values are 29% and 62% respectively. The authors must correct the values used.

The abstract states that the study demonstrates the promotion of biodiversity conservation.

Line 81. References to be inserted in brackets.

Lines 87, 89, 92. Research questions should end with question marks.

The figures in the papers are in reverse order with Figure 1 appearing as the last figure.

Line 249. Period missing between costs and Total costs.

Line 275 meat consumption is expected to fall and not deteriorate.

Lines 352-354 are repeated in line 359-361.

Line 400 is incomplete.

Line 424 corroborate should be followed by ‘with’.

**Do you want your identity to be public for this peer review?** For information about this choice, including consent withdrawal, please see our Privacy Policy

Reviewer #1: No

Reviewer #2: No

---

## [Author Response · Author response to Decision Letter 1]

24 Dec 2025

We would like to thank the Editor and the Reviewers for their careful reading of our manuscript and for their constructive and insightful comments. We greatly appreciate the time and effort invested in evaluating our work. The comments have helped us to clarify the novelty of the paper, strengthen the methodological transparency of the review, improve the consistency and precision of terminology, and sharpen the policy relevance of the analysis, particularly in the Greek context.

We have carefully considered all major and minor comments and revised the manuscript accordingly. In doing so, we have expanded and refined the introduction to better articulate the contribution of the review, clarified the rationale for model selection and evaluation criteria, corrected and calibrated specific model descriptions, and improved the overall structure, readability, and presentation of results. Below, we provide a detailed, point-by-point response to each comment, indicating how and where the manuscript has been revised.

Main Comments

1. The dataset summary is missing in the text. The reader therefore cannot make sense of what data was used for the analysis in the FABLE calculator.

Following a very helpful exchange with the Journal we have included the data required to produce the results and the graphs in the manuscript (based on projections with the FABLE Calculator) in the form of Supporting Information files uploaded to the submission system and explicitly stated “Data was generated for this study using the FABLE Calculator, which is publicly available via the following URL (Zenodo): https://zenodo.org/records/14638582.

Additional data, including a projection of the variables and other relevant information, have been provided in the form of Supporting Information files”.

2. It is suggested that the author include in the introduction the relevant research of other scholars on the combination of technological innovation and sustainability in the Greek agricultural sector and summarize and analyze the innovation and advantages of this study compared with it.

We have enhanced the introduction in the revised manuscript to include the work of other scholars in the Greek context (pp. 2-3). Despite the limited contributions on this specific topic, we include significant contributions and underscore the integration of the theoretical hypotheses with the data-driven projections from the FABLE Calculator, which is unique to our study. Furthermore, we complement the policy recommendations with the evidence from the projections to substantially contribute to the existing literature (p. 4).

3. In section 3.1, it is suggested that the author divide the relevant description of the productivity improvement path and the relevant results of the productivity improvement path into two parts for discussion, such as adding corresponding subheadings respectively, to make the content of this part clearer.

Section 3.1 has been explicitly divided into 2 subsections, namely “Construction of Pathways” and “Results”. This improves readability, with the first subsection preparing the ground for and facilitating the interpretation of the results (the FABLE projections for the key variables) described in the second subsection. (pp. 7-9)

4. In Section 3.2, the author compares and analyzes the results of national commitment scenario and BAU scenario in AFOLU emission and pastureland but lacks quantitative analysis of relevant results.

In the revised version of the paper, we have enhanced the discussion on AFOLU emissions and pastureland (hectares) for the comparison between the National Commitments and Current Trends Pathways in pp. 12-13. Furthermore, we explicitly refer to the components of GHG emissions in the crop and livestock sector (by pollutant, namely CH4 and N2O) to enhance the granularity of the analysis in both Sections 3.1 and 3.2.

5. In the third part, the author discussed the differences in results under different paths, but the reasons for this lack of more in-depth demonstration, such as the support of relevant literature. It is suggested that the author supplement this.

In both Sections 3.1 and 3.2 we complement the analysis of the results (projections) with a comparison with the relevant literature for the Greek case and beyond. We distinguish between corroborating findings in the literature for AFOLU GHG emissions (p.10 and p.13), land use and agricultural production costs (p.11 and p.15). Overall, this enhances the validity and relevance of our results, as the hypotheses are supported by an array of studies in literature.

6. When the author discusses horizontal policy and vertical policy in the fourth part, he should start from the perspective of the results of various indicators under the various paths of this paper, and there is no correlation between the two.

The first introductory paragraph in Section 4 (p. 16), before diving into the horizontal and vertical policies to support productivity growth, links the discussion with the results presented in Section 3 and underlines the main pillars of these policies. Furthermore, within sections 4.1 and 4.2 we have identified the correlation between the two policy sets and the implications they have for key variables as expressed in the FABLE C projections in Section 3 (p. 17 for horizontal policies and pp. 18-19 for vertical policies).

7. It is suggested that the author add the relevant discussion on the shortcomings of this study and the research issues that can be further explored in the future in the summary of Section 6

A paragraph of the limitation of the current research has been added to the Conclusion (p.24)

Specific Comments

1. In the Abstract, the figures for reduction of GHG emissions are 21% until 2030 and 52 % until 2050. However, in the text, the values are 29% and 62% respectively. The authors must correct the values used.

All values and statements in the Abstract are corrected in the revised manuscript.

2. The abstract states that the study demonstrates the promotion of biodiversity conservation

We have revised the Abstract accordingly and it now summarizes the key outputs included in the Results section

3. Line 81. References to be inserted in brackets.

All references in text are in brackets in the revised manuscript

4. Lines 87, 89, 92. Research questions should end with question marks.

Question marks have been inserted

5. The figures in the papers are in reverse order with Figure 1 appearing as the last figure.

The order of the figures has been corrected for consistency

6. Line 249. Period missing between costs and Total costs.

Period inserted

7. Line 275 meat consumption is expected to fall and not deteriorate.

Wording corrected

8. Lines 352-354 are repeated in line 359-361.

Duplicate lines have been removed

9. Line 400 is incomplete.

All typos and ambiguities in text have been corrected in the revised manuscript

10. Line 424 corroborate should be followed by ‘with’.

Wording has now been corrected

---

## [Editor Report · Decision Letter 1]

14 Jan 2026

The multi-faceted effects of technology-driven productivity surge in the crop & livestock sector in Greece: Evidence from the FABLE Calculator

PONE-D-24-52271R1

Dear Dr. Konstantinos Dellis,

We’re pleased to inform you that your manuscript has been judged scientifically suitable for publication and will be formally accepted for publication once it meets all outstanding technical requirements.

Kind regards,

Federico Zilia

Academic Editor

PLOS One

---

## [Editor Report · Acceptance letter]

PONE-D-24-52271R1

PLOS One

Dear Dr. Dellis,

I'm pleased to inform you that your manuscript has been deemed suitable for publication in PLOS One. Congratulations! Your manuscript is now being handed over to our production team.

Kind regards,

on behalf of

Dr. Federico Zilia

Academic Editor

PLOS One